# Wheat Crop Yield and Changes in Soil Biological and Heavy Metals Status in a Sandy Soil Amended with Biochar and Irrigated with Drainage Water

**Mohieyeddin M. Abd El-Azeim** [1], **Ahmad M. Menesi** [1], **Mahmoud M. Abd El-Mageed** [2], **Joanna Lemanowicz** [3] and **Samir A. Haddad** [4,*]

1   Soil Science Department, Faculty of Agriculture, Minia University, El-Minia 61519, Egypt
2   Agronomy Department, Faculty of Agriculture, Minia University, El-Minia 61519, Egypt
3   Department of Biogeochemistry and Soil Science, Bydgoszcz University of Science and Technology, 85-029 Bydgoszcz, Poland
4   Department of Agricultural Microbiology, Minia University, El-Minia 61517, Egypt
*   Correspondence: samir.mohamed@mu.edu.eg

**Abstract:** The current research aims to study the impacts of adding corncob biochar to a sandy soil irrigated with drainage water on wheat productivity, heavy metals fate, and some soil properties that reflect healthy soil conditions. This research consists of two separate experiments under field (lysimeters) and pot incubation conditions conducted on sandy soil irrigated with drainage water and treated with corncob biochar at the rate of 0.0, 1, 2, and 3% as mixing or mulching. Results specified that drainage water electrical conductivity value (5.89 dS m$^{-1}$) lies under the degree of restriction on use of "Severe", indicating that nonstop irrigation with such drainage water may cause a severe salinity problem in soil in the long run. A comparison of heavy metal concentrations of biochar-treated soils with the control showed that total heavy metals had accumulated significantly in the topsoil layer. Most of the available heavy metal concentrations in all soil leachate fractions were below the method detection limits. Mean concentrations of Ni, Cd, and Pb in wheat crops were far below the concentrations considered phytotoxic to wheat plants. More than 90% of the Ni, Cd, and Pb contained in the drainage water of the Al-Moheet drain were significantly present ($p \leq 0.05$) and adsorbed by biochar in the top 20 cm of soil lysimeters, indicating the high biochar adsorptive capacity of heavy metals. Total counts of bacteria and fungi gradually and significantly increased over the soil incubation time despite irrigation with contaminated drainage water. Soil resistance index (SRI) values for microbial biomass were positive throughout the experiment and increased significantly as the application rate of corncob biochar increased. These results indicated the high feasibility of using corncob biochar at a rate of 3% to temporarily improve the health of sandy soil despite irrigation with drainage water.

**Keywords:** soil properties; wheat productivity; corncob biochar; drainage water; microbial biomass

## 1. Introduction

The different farming systems in Egypt consume approximately 85% of the entire water budget of the Nile River, limited rainfall, groundwater, and non-conventional water resources. Surface flood irrigation is the predominant irrigation system using high- and low-quality water in the ancient Nile Valley and Delta lands, with less than 50% utilization efficiency, causing pollution and huge losses from these valuable water resources [1,2]. In Egypt, the scarcity of water is initially physical scarcity because of limited water resources, and furthermore, economic scarcity as a result of using contaminated water and improper management of water resources. Accordingly, Egypt faces enormous issues due to insufficient available water resources, represented by its fixed share of the Nile water, which amounts to 55.5 billion cubic meters annually.

In the future, the food gap in Egypt will be severe due to projected rapid population growth and water scarcity because of the construction of the Grand Ethiopian Renaissance Dam (GERD) and possibly due to rapid climate change [2,3]. Thus, food production must increase to meet population growth as feeding Egypt's population of 104 million as of 2021 exceeds the requirements for restoring and improving arable soil and water quality and quantity [4]. With the increasing population pressure and food gap in Egypt, the need for additional soil reclamation and irrigation water is forcing the country to use all non-conventional resources of low-quality water such as drainage water, brackish and saline water, wastewater, and even seawater in the future [1,5]. There is an urgent need for a sustainable agricultural system and technology, for example, biochar-treated water and soil, that can help increase soil productivity and reduce water pollution [6,7].

Under arid conditions, agricultural production is one of the chief elements that contribute to economic growth and food security, despite the accompanying difficulties such as lack of water, low soil fertility, desertification, salinity, and low crop yield. These difficulties can be relatively relieved by using biochar treatment technology for irrigation water and cultivated soils. This technology has become the focus of water treatment research as, compared to other physical and chemical methods, it provides environmental purity, health safety, and ease of use. All the research conducted over the last decades has indicated the positive role of biochar application in agricultural ecosystems to increase the quality of irrigation water and soil nutrients, reduce heavy metal availability in soil, and increase crop yield [4,5,7,8].

Biochar is a porous structured solid product, obtained when biomass is thermochemically treated in an oxygen-limited environment. Biochar has been applied in agriculture to different soils as a new carbon rich-material to improve soil health and for heavy metal remediation [9–12]. Soil pollution from heavy and trace metals because of irrigation with contaminated drainage water has warranted a stern warning because of their non-degradable nature and bio-availability to plant and soil organisms, thus heavy metals are susceptible to entering the food chain [7,13]. The accretion of heavy and trace metals in soils such as Ni, Pb, and Cd is of increasing concern owing to their fatal, cancer-causing, and multiple dysfunctional effects on organisms [14].

Biochar treatment of soils and irrigation water for remediation has been studied and adopted worldwide in the agriculture sector [15,16]. Considering biochar's widespread use around the world, using it to treat soil or water has received much less attention. Consequently, the present work aimed to study the effects of biochar addition on sandy soil properties, the fate of heavy metals, wheat crop yield, and its application for sustainable agricultural practices under arid conditions. To achieve this aim, the objectives of the present work were (i) to evaluate the irrigation suitability of Al-Moheet drainage water; (ii) to evaluate drainage water effects upon wheat crop yield and quality parameters; (iii) to evaluate the effects of corncob biochar-treated soil irrigated with Al-Moheet drainage water upon heavy metals fate; and (iv) to evaluate effects of corncob biochar on sandy soil's biological properties irrigated with Al-Moheet drainage water.

In addition, another motivation of the present research was to study how corncob biochar affects soil microbial biomass and enhances soil enzymatic activity in sandy soil irrigated with drainage water contaminated with heavy metals. Such information is required to ensure secure and healthy soil and agricultural products in newly reclaimed sandy soils irrigated with drainage water under field conditions of arid regions.

## 2. Materials and Methods

To achieve the objectives of this research, two separate experiments were conducted. The first was a crop lysimeters experiment for wheat yield and quality parameters at a private farm in the Shosha district, El-Minia Governorate, Egypt, and the second was a soil pot incubation experiment for biological parameters at the facilities of the Agricultural Microbiology Department, Faculty of Agriculture, Minia University, Egypt. El-Minia Governorate, Egypt is located at 28°07′28″ N 30°44′03″ E and has an arid climate with

hot, dry summers and cold winters; the average maximum and minimum temperature is 30.4 degrees Celsius and 44.1 °C in summer and 1.6 °C and 23.7 °C in winter, respectively, with wide daily differences. The average annual precipitation is almost 25 mm due to severe climate changes, and approximately 90 percent of these are usually received within a short period of two months, in January and February.

### 2.1. Wheat Crop Lysimeters Experiment

2.1.1. Experimental Materials and Design

To achieve the objectives of this research, two separate experiments were conducted, the first was a crop lysimeter experiment for wheat yield Giza 171 (*Triticum aestivum* L.) and quality parameters at a private farm, Shosha district, El-Minia Governorate, Egypt, and the second was a soil incubation experiment for soil biological parameters at the facilities of Faculty of Agriculture, Minia University, Egypt. Lysimeters were established in newly reclaimed sandy soil, filled with sandy soil collected from the top 0–30 cm layer after the soil was air-dried and sieved through a 2 mm sieve. These newly reclaimed sandy soil fields were first cultivated 5 years ago with an Egyptian clover, wheat, barley, and maize rotation regime. Maize waste of corncobs was obtained after maize harvest, and corncob biochar was prepared by pyrolysis at 300 °C in the absence of oxygen [17].

Corncob biochar was air dried and ground to a 2 mm size before use in lysimeter trials. Actual amounts of corncob biochar added to the top of 15 cm of each lysimeter were equal to application rates of 0 (control), 1, 2, and 3%, respectively. Sub samples of biochar were scanned using a sophisticated Scanning Electron microscope (SEM), Joel Model SEM 5910. Biochar scanned images were captured at different magnification levels to signify biochar morphology (Figure 1).

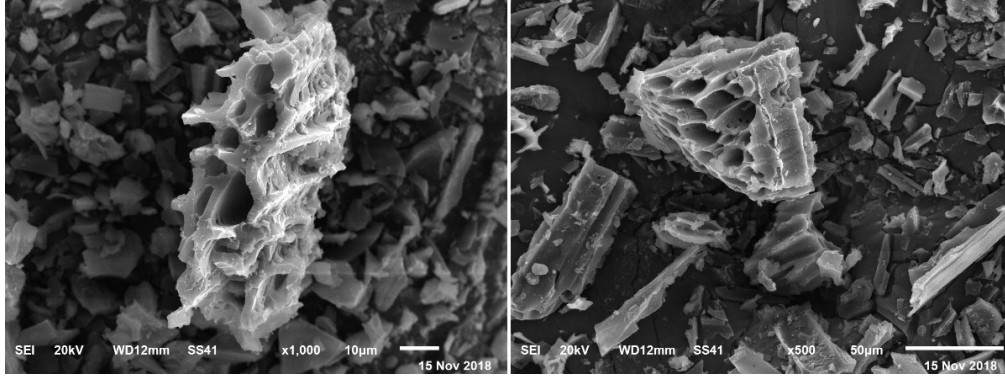

**Figure 1.** Image of scanning electron microscope of corncob biochar used in the lysimeters experiment.

The physical and chemical properties of biochar and sandy soil were investigated using soil sub-samples and 2 mm sized biochar particles [18,19] in the Soil Survey Laboratory [20]. Some important characteristics of the investigated sandy soil and corncob biochar are given in Tables 1 and 2.

**Table 1.** Physiochemical characteristics of the investigated soil.

| Soil Property | |
|---|---|
| Sand (%) | 90.30 |
| Silt (%) | 7.14 |
| Clay (%) | 2.56 |
| Texture grade | Sand |
| F.C [%] | 14.98 |
| PWP [%] | 4.45 |
| WHC [%] | 19.22 |
| AV. W (F.C—PWP) [%] | 10.53 |

**Table 1.** *Cont.*

| Soil Property | |
|---|---|
| BD g cm$^3$ | 1.63 |
| PD g cm$^3$ | 2.61 |
| pH in H$_2$O | 8.59 (8.31) * |
| CEC [cmolc kg$^{-1}$ soil] | 3.60 |
| ECe [dS m$^{-1}$] | 1.56 |
| OM [g kg$^{-1}$] | 2.8 ** |
| SOC [g kg$^{-1}$] | 1.62 |
| TN [g kg$^{-1}$] | 0.15 |
| SOC/TN | 10.80 |
| TP [g kg$^{-1}$] | 0.09 |
| TK [g kg$^{-1}$] | 1.7 |
| TNi [mg kg$^{-1}$] | 46.20 |
| TPb [mg kg$^{-1}$] | 57.00 |
| TCd [mg kg$^{-1}$] | 4.10 |

* Numbers in brackets are pH values obtained for soil by Ca Cl$_2$ ratio of 1:2.5; ** Organic matter determined by loss on ignition. F.C—field capacity, PWP—permanent wilting point, WHC—water holding capacity, BD—bulk density; CEC—cation exchange capacity; EC—electrical conductivity; OM—organic matter; SOC—soil organic carbon; TN—total nitrogen; TP—total phosphorus, TK—total potassium; TNi—total nickel; TPb—total lead; TCd—total cadmium.

**Table 2.** Selected physiochemical characteristics of the studied corncob biochar.

| Biochar Property | |
|---|---|
| BD [g cm$^3$] * | 0.29 |
| WHC [%] | 72.78 |
| pH in H$_2$O | 6.38 |
| EC [dS m$^{-1}$] | 0.651 |
| CEC [cmol(+) kg$^{-1}$ soil] | 54.44 |
| Ash [%] | 8.59 |
| TOC [g kg$^{-1}$] | 368 |
| TN [g kg$^{-1}$] | 17.80 |
| TOC/TN | 20.67 |
| TP [g kg$^{-1}$] | 3.5 |
| TN/TP | 5.08 |
| K$^+$ [mg kg$^{-1}$] | 483.7 |
| Ca$^{2+}$ [mg kg$^{-1}$] | 668.9 |
| Mg$^{2+}$ [mg kg$^{-1}$] | 142.7 |
| TNi [mg kg$^{-1}$] | 15.89 |
| TPb [mg kg$^{-1}$] | 11.87 |
| TCd [mg kg$^{-1}$] | 0.76 |

* Abbreviations: See Table 1.

2.1.2. Lysimeter Experiment Construction

To study the effects of corncob biochar addition to sandy soil irrigated with drainage water on the accumulation, leachability, and bioavailability of heavy metals for wheat plants, twenty-four lysimeters (8 treatments × 3 replicates) with an inner diameter of 1.0 m and 1.5 m depth were constructed under field conditions in accordance with the illustration and design in Figure 2.

Before treating the soil in lysimeters with biochar, each lysimeter base was filled with a 10 cm layer of gravel to form a free-draining base, then the investigated sandy soil was placed on top of the free-draining base material to a settled depth of 1.4 m. Corncob biochar was then added and incorporated by hand into 20 cm topsoil of each lysimeter or as mulch over the lysimeter soil surface. To prove the effects of intensive flood irrigation with drainage water of corncob biochar-treated sandy soil on the fate of heavy metals (Ni$^{2+}$, Cd$^{2+}$ and Pb$^{2+}$), samples of similar-sized wheat plants and grains were taken from each lysimeter directly before harvest. Furthermore, soil samples were collected and analyzed for heavy

metals to quantify to what degree the heavy metals ($Ni^{2+}$), ($Cd^{2+}$), and ($Pb^{2+}$), applied via irrigation with drainage water to a coarse-textured sandy soil, have accumulated or moved into the soil profile in the presence of corncob biochar.

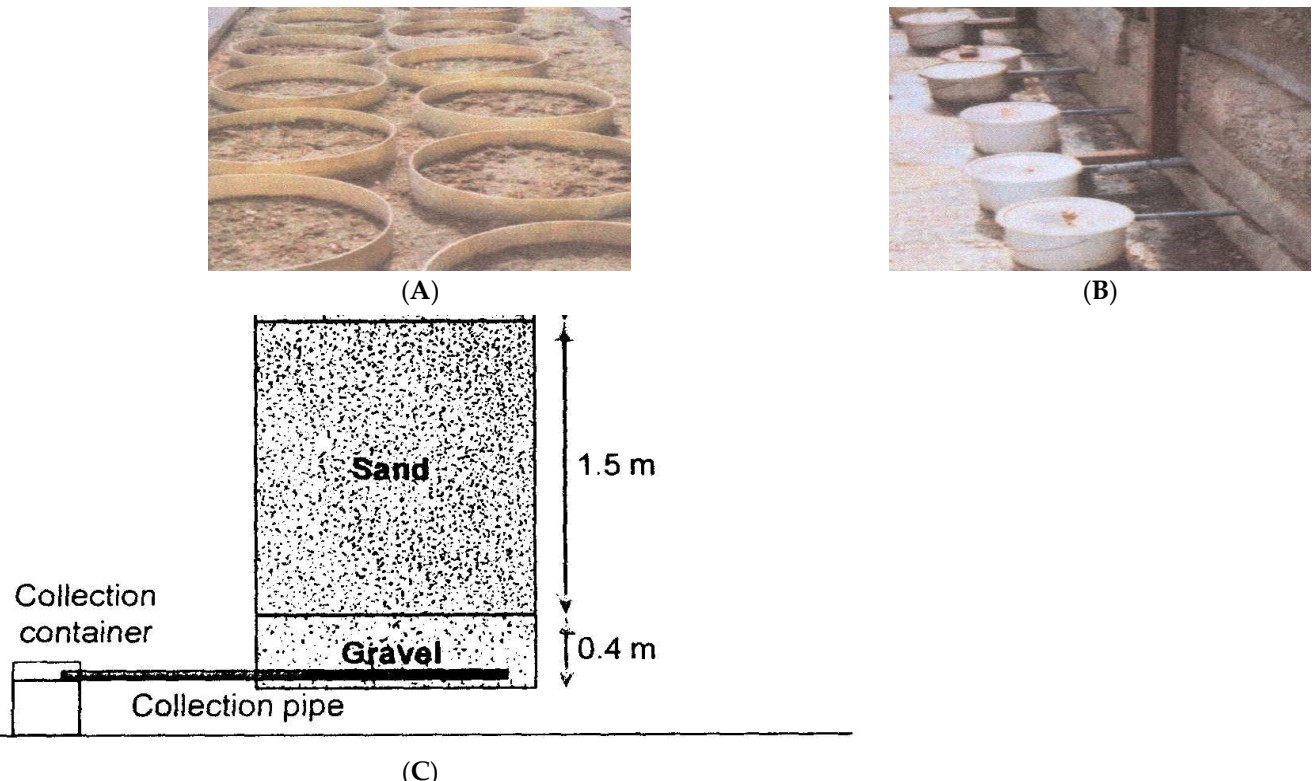

**Figure 2.** Lysimeter experiment construction. (**A**) Experimental lysimeters view. (**B**) Leachate collection buckets at the base of each lysimeter. (**C**) Schematic diagram of the lysimeter side elevation.

Analyses of heavy metals content in wheat plant tissues were determined via digestion in boiling aqua regia prepared in accordance with ISO 11466 [21], using Electro-Thermal Atomic Absorption Spectrometry [22]. Leachate samples were collected every 25 days after 7 irrigation events over a 160-day period of wheat cultivation. Each sample was filtered in the laboratory with Whatman no. 2 filter paper and analyzed for pH and extracted for the analysis of total concentrations of nonessential heavy metal Ni, Cd, and Pb using Electro-Thermal Atomic Absorption Spectrometry [22]. Analyses of heavy metals content in corncob biochar and biochar-treated soils were determined by digestion in boiling aqua regia prepared in accordance with ISO 11466 [21], using Electro-Thermal Atomic Absorption Spectrometry [22]. The method detection limits for the selected heavy metals of Ni, Cd, and Pb are 0.01, 0.01, and 0.005 mg $L^{-1}$, respectively [23,24].

2.1.3. Evaluation of Al-Moheet Drainage Water Quality for Irrigation

Al-Moheet drain is the main drain in El-Minia Governorate and extends 135 km along the length of the governorate, and all types of agricultural and industrial waste and sewage are drained in this drain. Unfortunately, farmers in Egypt still use this drain's water for irrigation (Figure 3). To evaluate water suitability for wheat irrigation, water samples of collected drainage water were analyzed for its chemical composition. In each irrigation event, irrigation water was collected on the same irrigation day and used directly for irrigation in the morning. An irrigation water sub-sample was collected from the Al-Moheet drain in a clean and dried plastic beg, filtered, and analyzed in accordance with the American Public Health Association [25]. The chemical composition of Al-Moheet drainage water used for the irrigation of wheat plants is presented in Table 3.

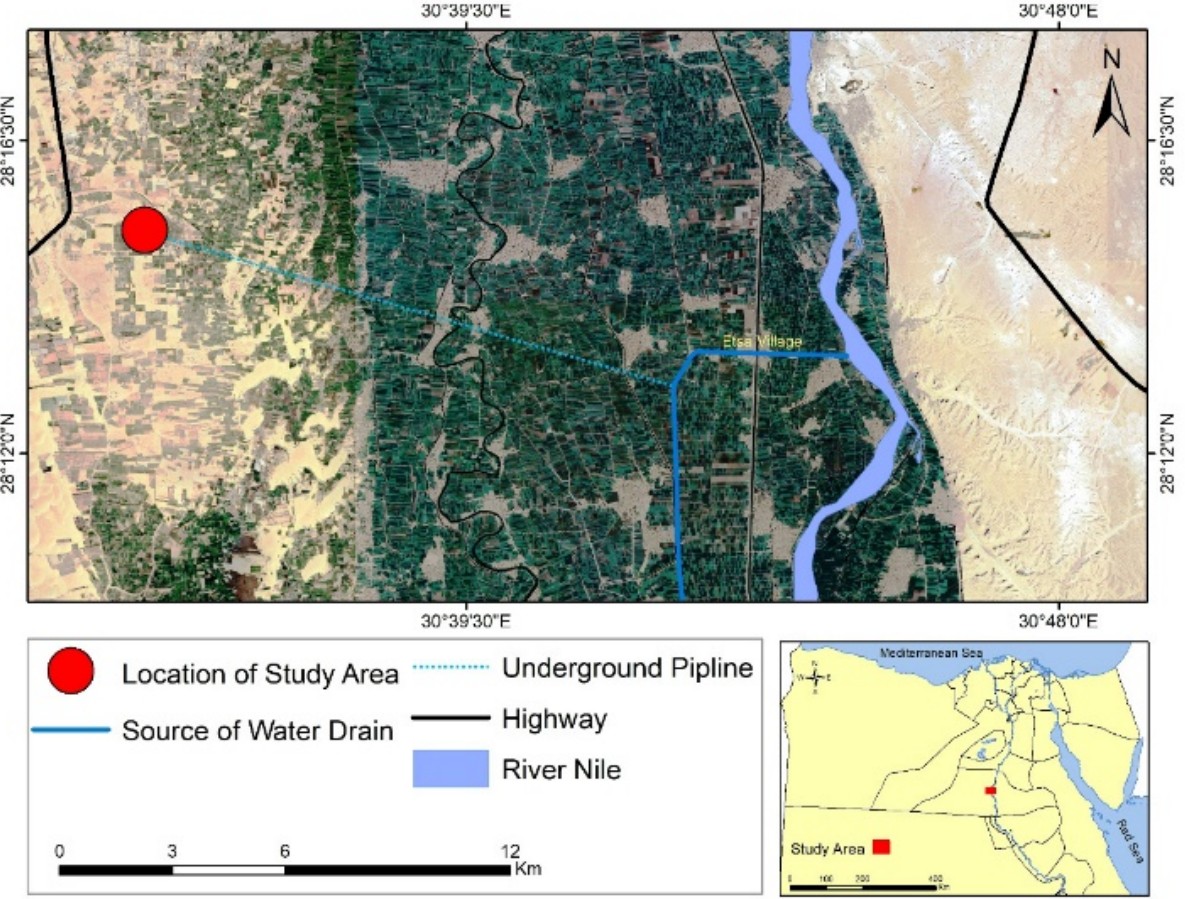

**Figure 3.** Location map showing Al-Moheet drain and study area.

**Table 3.** Water chemical composition of al-Moheet drain water.

| Chemical Composition | |
|---|---|
| pH | 8.89 |
| EC [dS m$^{-1}$] | 5.89 |
| TDS [mg L$^{-1}$] | 4712 |
| Soluble Ca$^{2+}$ [mmolc L$^{-1}$] | 40.54 |
| Soluble Mg$^{2+}$ [mmolc L$^{-1}$] | 21.82 |
| Soluble Na$^+$ [mmolc L$^{-1}$] | 25.73 |
| Soluble K$^+$ [mmolc L$^{-1}$] | 2.67 |
| Soluble Cl$^-$ [mmolc L$^{-1}$] | 32.78 |
| Soluble SO$_4^{2-}$ [mmolc L$^{-1}$] | 41.89 |
| Soluble HCO$^{3-}$ + CO$_3^{2-}$ [mmolc L$^{-1}$] | 16.78 |
| **Water Heavy Metals Content** | |
| Ni [mg L$^{-1}$] | 120 |
| Pb [mg L$^{-1}$] | 75 |
| Cd [mg L$^{-1}$] | 3.6 |
| **Chemical Criteria** | |
| SAR | 4.61 |
| Ca$^{2+}$/Mg$^{2+}$ | 1.86 |
| MH [%] | 34.99 |
| Na$^+$/Cl$^-$ | 0.78 |
| Na [%] | 31.39 |
| RSC Residual Sodium Carbonate | <1.25 |

SAR—Sodium Adsorption Ratio; MH—Magnesium Hazard; RSC—Residual Sodium Carbonate.

## 2.2. Incubation Experiment

After wheat harvest, to study the effects of corncob biochar-treated sandy soil irrigated with drainage water on soil microbial and enzyme activity, two kilograms of the sandy soil treated with corncob biochar was obtained from the top 0–20 cm lysimeter layer, then the soil was air-dried and incubated for 20 days under 25 °C. The soil water content was adjusted to the field capacity using Al-Moheet drainage water by calculating the weight differences of soil pots.

### 2.2.1. Soil Microbial Biomass

After incubation, the impacts of irrigation with Al-Moheet drainage water on soil microbial biomass in sandy soil were determined. Ten grams was taken from incubated sandy soils and added to 95 mL of sterile water, which was agitated for 5 min, and then the solution was diluted ($10^{-1}$ to $10^{-6}$) and the resulting solutions were inoculated on the nutrient agar medium and Martin's [26] medium for bacteria and fungi, respectively. Colony forming units (CFU) of bacteria ($\times 10^6$ cfu g$^{-1}$) and fungi ($\times 10^4$ cfu g$^{-1}$) were counted every five days during the course of the incubation period.

### 2.2.2. Soil Resistance Index (SRI)

The soil resistance index (SRI) was determined using the following equation by Orwin and Wardle [27] using the total count of bacteria and fungi that resisted heavy metal contamination of soil through irrigation with Al-Moheet drainage water.

$$\text{SRI} = 1 - \frac{2\,|D_0|}{(C_0 + |D_0|)} \tag{1}$$

where $D_0$ is the difference between the untreated control ($C_0$) and soil samples treated with biochar ($P_0$). SRI values range between $-1$ and $+1$ with a value of $+1$ indicating that the disorder has no effect (higher resistance) and lower values showing a stronger opposite effect (lower resistance).

### 2.2.3. Sandy Soil Enzymatic Activity

All enzyme activities were expressed as products per unit of dry soil mass and incubation time. Urease (UR) activity was evaluated in soil samples using urea as a substrate via spectrophotometry at 578 nm according to Tabatabai [28]. On the other hand, the activity of arginase (AR) was determined as described by Alef and Kleiner [29]. Arginase activity in the soil extract was measured via spectrophotometry at 630 nm.

## 2.3. Statistical Analyses

The experimental data were subjected to an analysis of variance using a completely randomized block design (CRBD) with three replicates [30]. The significance of the differences was estimated and compared using the Duncan test at a 5% level of probability ($p < 0.05$).

## 3. Results and Discussion

### 3.1. Wheat Crop Lysimeters Experiment

The results obtained from this experiment for wheat grown in biochar-amended sandy soil irrigated with Al-Moheet drainage water are presented under the following sub-items.

### 3.1.1. Evaluation of Al-Moheet Drainage Water Quality for Irrigation

Regarding water classification by salinity in accordance with Ayers and Westcott [31], Al-Moheet drainage water is classified as severely saline water as its electrical conductivity value (5.89 dS m$^{-1}$ at 25 °C) lies under the degree of severe restriction on use, indicating that using such drainage water in irrigation may cause a severe salinity problem in the soil in the long run (Table 3).

The obtained results indicated that Al-Moheet drain water is unsuitable for wheat crop irrigation in terms of certain chemical water quality criteria such as water pH, the Na/Cl ratio, the Ca/Mg ratio, magnesium hazard, and high concentrations of total dissolved salts (4712 mg $L^{-1}$), EC (5.89 dS $m^{-1}$), chloride (32.78 mmol$_c$ $L^{-1}$), and bicarbonate (16.78 mmol$_c$ $L^{-1}$). On the other hand, drainage water was suitable for irrigation in terms of residual sodium carbonate (RSC < 0.25). Higher levels of water salinity with intensive irrigation significantly increased the initial soil electrical conductivity, soil salinity build up, and soil pH despite the presence of corncob biochar. As a result of physicochemical Al-Moheet drainage water characteristics, this drainage water may be used carefully for irrigation of some suitable crops only under certain conditions (Table 3).

### 3.1.2. Combined Impacts of Corncob Biochar and Drainage Water on Wheat Productivity and Quality Characteristics

The data in Table 4 show that the studied wheat productivity and quality parameters were significantly affected by biochar addition rates, where the highest values were recorded under 3% biochar treatment compared with 1% and 2%. biochar treatments In general, wheat productivity and quality parameters were significantly affected by biochar addition at all application rates compared to control.

**Table 4.** Impacts of corncob biochar on wheat productivity and quality parameters.

| Treatment Rate | Spike Length [cm] | 1000-Grain Weight [g] | Grain Yield [t ha$^{-1}$] | Straw Yield [t ha$^{-1}$] | Biological Yield [t ha$^{-1}$] | Harvest Index [%] |
|---|---|---|---|---|---|---|
| Biochar Incorporated | | | | | | |
| 0% | 9.55 [b*] | 40.13 [b] | 4.75 [d] | 6.63 [c] | 11.38 [c] | 41.74 |
| 1% | 8.99 [b] | 39.06 [b] | 4.89 [c] | 6.85 [bc] | 11.74 [bc] | 41.65 |
| 2% | 9.26 [b] | 37.45 [b] | 6.22 [b] | 8.17 [b] | 14.39 [b] | 43.24 |
| 3% | 10.69 [a] | 43.28 [a] | 6.94 [a] | 8.81 [a] | 15.75 [a] | 44.06 |
| Biochar Mulching | | | | | | |
| 0% | 9.55 [b] | 40.13 [b] | 4.75 [d] | 6.63 [c] | 11.38 [c] | 41.74 |
| 1% | 8.99 [b] | 39.06 [b] | 4.92 [c] | 6.91 [bc] | 11.83 [bc] | 41.59 |
| 2% | 9.26 [b] | 37.45 [b] | 6.35 [b] | 8.26 [b] | 14.61 [b] | 43.46 |
| 3% | 10.69 [a] | 43.28 [a] | 6.91 [a] | 8.97 [a] | 15.88 [a] | 43.51 |

* Means in each column appointed by the same letter are not significantly different at 5% level.

The increase in wheat yield and quality parameters at the biochar rate of 3% may be due to the fact that the amount of water applied was the optimum amount for good plant growth, avoiding water stress conditions on control treatment due to the high infiltration rate of the investigated sandy soil, and therefore water stress badly affects plant growth and expansion of plant cells.

Corncob biochar incorporated into or applied to the surface of sandy soil significantly increased certain soil chemical properties such as pH, CEC, OM, EC, and total N compared to the control (Table 5). This has a positive effect where nutrients might be more available in such alkaline soils. Similar pronounced and significant improvements in certain sandy soil physical properties, such as decreasing the bulk density and increasing water moisture constants, were observed at all application rates of corncob biochar compared to the control.

Improvements in soil physicochemical characteristics consequently influence the soil microbial activity, soil fertility, and bioavailability of heavy metals in wheat crops. The application of corncob biochar to sandy soil is expected to affect the soil moisture, pH, salinity, OM, and CEC, which, in turn, could influence the microbial biomass and fate of heavy metals bioavailability in the amended soil [5,32,33].

**Table 5.** Selected sandy soil properties changed after corncob biochar application.

| Biochar Rate | Soil Chemical Properties | | | | Soil Particle Size % | | | | Soil Moisture % | | | |
|---|---|---|---|---|---|---|---|---|---|---|---|---|
| | TN | pH | CEC | EC | OM | Sand | Silt | Clay | WHC | FC | PWP | AV. W |
| 0% | 0.015 c* | 8.59 a | 3.60 d | 1.56 c | 0.28 d | 90.30 a | 7.14 d | 2.56 d | 19.22 d | 14.98 d | 4.45 a | 10.53 d |
| 1% | 0.05 b | 8.31 b | 6.60 c | 2.08 b | 0.58 c | 87.00 b | 9.50 a | 3.50 c | 21.4 c | 16.35 c | 4.10 c | 12.25 c |
| 2% | 0.06 b | 8.31 b | 8.80 b | 2.11 b | 0.66 b | 85.30 c | 8.10 c | 6.60 a | 23.12 b | 18.22 b | 4.30 b | 13.92 b |
| 3% | 0.08 a | 8.22 c | 11.30 a | 2.36 a | 0.89 a | 85.10 d | 8.60 b | 6.30 b | 25.14 a | 19.35 a | 4.50 a | 14.85 a |

* Figures in the same column followed by same letters are not significant. Figures represent means of incorporation plus mulching since there was no significant differences between both methods of biochar application. TN—total nitrogen [%], pH soil reaction, CEC—cation exchange capacity [cmol(+) kg], EC—electrical conductivity [dS m$^{-1}$], OM—organic matter [%], WHC—water holding capacity, FC—field capacity, PWP—permanent wilting point, AV.W—available water.

### 3.1.3. Combined Impacts of Corncob Biochar and Drainage Water on Lysimeters Soil Leachate Composition

$Ni^{2+}$, $Cd^{2+}$, and $Pb^{2+}$ movement to the lower soil profile (0 to 150 cm) in wheat crop sandy soil lysimeters was evaluated after biochar application (Table 6). Data from leachate analyses show wide assurance about the heavy metal concentrations in leaching water in terms of environmental protection (Table 6). Most leachate fraction concentrations of Ni, Pb, and Cd collected from untreated lysimeters (control) increased gradually to hazardous levels throughout the study after all irrigation events compared to lysimeters treated with biochar. This could be attributed to the absence of biochar as an organic adsorbent material.

**Table 6.** Concentrations of Ni, Pb, and Cd collected from lysimeters' leachate.

| Element | Biochar Rate | Elements Concentrations in Leachate Fractions mg L$^{-1}$ | | | | | | | | | | | |
|---|---|---|---|---|---|---|---|---|---|---|---|---|---|
| | | 25 Days | | 50 Days | | 75 Days | | 100 Days | | 125 Days | | 150 Days | |
| | | Inc. | Mul. | Inc. | Mul. | Inc. | Mul. | Inc. | Mul. | Inc. | Mul. | Inc. | Mul. |
| Ni | 0% | 1.08 * | 1.08 | 4.0 a | 4.06 a | 3.05 a | 3.05 a | 5.05 a | 5.05 a | 5.06 a | 4.42 a | 6.09 a | 6.07 a |
| | 1% | 0.17 | 0.13 | 0.26 b | 0.17 b | 0.26 b | 0.07 b | 0.06 b | 0.07 b | 0.05 b | 1.5 b | 1.56 b | 1.90 b |
| | 2% | UMDL | UMDL | 0.16 b | 0.18 b | 0.16 b | 0.07 b | 0.06 b | 0.08 b | 0.08 b | 0.06 b | 0.46 b | 0.88 b |
| | 3% | UMDL | UMDL | 0.06 b | 0.08 b | 0.07 b | 0.07 b | 0.07 b | 0.07 b | 0.07 b | 0.06 b | 1.61 b | 0.78 b |
| Pb | 0% | 1.08 | 1.05 | 2.07 a | 2.07 a | 0.74 a | 0.74 a | 2.07 a | 2.07 a | 2.07 a | 2.07 a | 4.78 a | 4.11 a |
| | 1% | UMDL | UMDL | 0.08 b | 0.08 b | 0.09 b | 0.09 b | 0.09 b | 0.08 bc | 0.08 b | 0.08 b | 1.45 bc | 2.9 ab |
| | 2% | UMDL | UMDL | 0.08 b | 0.09 b | 0.08 b | 0.08 b | 0.09 b | 0.08 bc | 0.09 b | 0.09 b | 1.22 b | 1.09 bc |
| | 3% | UMDL | UMDL | 0.08 b | 0.08 b | 0.08 b | 0.08 b | 0.06 d | 0.07 cd | 0.08 b | 0.08 b | 1.07 bc | 0.89 c |
| Cd | 0% | 0.12 | UMDL | 0.17 a | 0.26 a | 0.17 a | 0.17 a | 0.17 a | 0.15 a | 0.17 a | 0.15 a | 0.67 a | 0.34 b |
| | 1% | 0.15 | UMDL | 0.06 c | 0.08 cb | 0.09 b | 0.09 b | 0.12 ab | 0.08 b | 0.08 b | 0.08 b | 0.19 c | 0.19 c |
| | 2% | UMDL | UMDL | 0.08 bc | 0.09 cb | 0.09 b | 0.09 b | 0.09 b | 0.08 b | 0.09 b | 0.08 b | 0.06 d | 0.05 d |
| | 3% | UMDL | UMDL | 0.08 bc | 0.08 bc | 0.08 b | 0.08 b | 0.08 b | 0.07 b | 0.08 b | 0.08 b | 0.05 d | 0.07 d |

* Figures in the same column followed by the same letters are not significant. UMDL—under method detection limits, Inc.—incorporation method, Mul.—Mulching method.

Most leachate fraction concentrations of Ni, Pb, and Cd collected from the lysimeters treated with biochar remained under hazardous limits, and many leachate samples were below the method detection limits throughout the study for all biochar rates and methods of application when compared to the control. Only in some leachate fractions (after 50, 75, 100, 125, and 150 days) was a small portion of Ni and Pb removed and only in early leachates when biochar was applied to the lysimeters. The Cd concentrations were nearly below the method detection limits in most soil leachate fractions treated with biochar after the first irrigation event. In the control treatment (untreated soils), all leachate fractions (after 50, 75, 100, and 125 days) had a significantly higher proportion of Ni (ranging from 4.05 to 6.09 mg L$^{-1}$) and Pb (ranging from 2.07 to 4.78 mg L$^{-1}$) due to their readily water-soluble nature.

Leachate concentrations of Ni, Pb, and Cd in biochar treatments were significantly lower than the control in the second, third, fourth, and fifth leachate samples. These initial decreases in Ni, Pb, and Cd solubility were presumably due to the higher retention capacity of biochar. In general, Cd concentrations in all leachate fractions before 50 days of wheat

cultivation were below the method detection limits from all biochar-amended lysimeters. There were no significant differences between biochar incorporation or subsurface application at all application rates. It is, however, reported that heavy metal solubility and mobility in biochar-treated soil depends on the biochar application rate and method of application, biochar pyrolysis method and biochar feedstock, biochar heavy metal content, biochar and soil adsorptive properties, and the formation of insoluble precipitates [5,16,33–35].

3.1.4. Combined Impacts of Corncob Biochar and Drainage Water on Soil Heavy Metals Accumulation and Movements

Table 7 shows the total and extractable heavy metals content in the upper 20-cm sandy soil lysimeters. The addition of biochar resulted in higher metal loadings for the 3% biochar rate compared to the control (from 48.27 to 59.6 kg ha$^{-1}$ for Ni, from 58.2 to 81.5 kg ha$^{-1}$ for Pb, and from 4.6 to 6.4 for Cd kg ha$^{-1}$). In terms of heavy metals accumulation and bioavailability, the application of corncob biochar is safe despite flood irrigation with contaminated Al-Moheet drainage water. Moreover, the addition of biochar resulted in significant increases in extractable heavy metals under investigation (Ni, Pb, and Cd).

**Table 7.** Total and extractable heavy metals in the upper 20 cm soil as affected by biochar addition and drainage water irrigation.

| Treatments Rate | Soil Total Metals [mg kg$^{-1}$] | | | Extractable Heavy Metals [mg kg$^{-1}$] | | |
|---|---|---|---|---|---|---|
| | Ni | Pb | Cd | Ni | Pb | Cd |
| | **Biochar Incorporated** | | | | | |
| 0% | 48.27 [d]* | 58.20 [d] | 4.60 [a] | 1.37 [b] | 1.53 [d] | 0.04 [d] |
| 1% | 51.30 [c] | 68.30 [c] | 4.90 [a] | 2.90 [a] | 3.73 [c] | 1.03 [ab] |
| 2% | 56.40 [ab] | 74.60 [b] | 5.40 [a] | 3.13 [a] | 4.87 [ab] | 1.03 [b] |
| 3% | 59.60 [a] | 81.50 [a] | 6.40 [a] | 3.27 [a] | 5.23 [a] | 1.05 [a] |
| | **Biochar Mulching** | | | | | |
| 0% | 46.20 [d] | 58.20 [d] | 4.60 [a] | 1.37 [b] | 1.53 [d] | 0.04 [d] |
| 1% | 48.27 [c] | 67.60 [c] | 4.30 [a] | 2.97 [a] | 3.93 [bc] | 0.10 [c] |
| 2% | 55.20 [b] | 75.10 [b] | 5.20 [a] | 3.13 [a] | 4.83 [ab] | 1.04 [ab] |
| 3% | 58.60 [ab] | 80.30 [a] | 5.90 [a] | 3.47 [a] | 5.03 [a] | 1.05 [ab] |

* Figures in the same column followed by same letters are not significant.

The results of this study revealed that biochar application caused insignificant increases ($p \leq 0.05$) in the surface accumulation (top 20 cm of soil) of total heavy metals at all application rates. There were no significant differences between the different methods of application. In general, more than 90% of the biochar-applied heavy metals and irrigation water were present in the top 20 cm of soil plots. Results of this study showed that no increases in metal accumulation were found below the 20-cm depth and Ni, Pb, and Cd were all found around mulched biochar boundaries and to the depth of biochar incorporation, indicating the high capacity of corncob biochar to adsorb heavy metals.

A gradual increase pattern with significant differences was noted for the extractable Ni, Pb, and Cd in the biochar-amended lysimeters. Extractable concentrations of Ni, Pb, and Cd in biochar-amended lysimeters (0–20 cm) did not correlate significantly with the total applied amounts of these heavy metals in biochar and drainage water used for irrigation. In most cases, there were no significant differences in extractable heavy metal concentrations between all the biochar treatments at all application rates and methods of application (Table 8).

Biochar application insignificantly caused slight up and down changes in the Cd, Ni, and Pb concentrations below the top 20 cm of soil (Table 8). These results agree with various studies that found leaching studies have shown little or no short-term movement of heavy metals through the soil as a result of biochar applications and soil properties of arid soils [36]. Because of the large surface area and high capacity of biochar to adsorb

heavy metals when mixed with soils, heavy metals movement through a soil profile is not expected [5,16].

**Table 8.** Total and extractable heavy metals under the upper 20 cm soil as affected by biochar addition and drainage water irrigation.

| Treatments Rate | Soil Total Metals [mg kg$^{-1}$] | | | Extractable Heavy Metals [mg kg$^{-1}$] | | |
|---|---|---|---|---|---|---|
| | Ni | Pb | Cd | Ni | Pb | Cd |
| **Biochar Incorporated** | | | | | | |
| 0% | 46.26 [a*] | 57.40 [b] | 4.20 [d] | 1.23 [c] | 1.40 [c] | 0.04 [b] |
| 1% | 46.60 [a] | 59.20 [a] | 4.20 [d] | 1.37 [bc] | 2.17 [a] | 0.04 [b] |
| 2% | 47.90 [a] | 60.30 [a] | 3.90 [e] | 1.67 [a] | 1.80 [b] | 0.06 [b] |
| 3% | 47.50 [a] | 60.30 [a] | 4.40 [b] | 1.47 [b] | 2.30 [a] | 0.08 [b] |
| **Biochar Mulching** | | | | | | |
| 0% | 46.26 [a] | 57.40 [b] | 4.20 [d] | 1.23 [c] | 1.47 [c] | 0.04 [b] |
| 1% | 47.10 [a] | 59.87 [a] | 4.30 [d] | 1.27 [c] | 1.83 [b] | 0.60 [a] |
| 2% | 47.50 [a] | 59.10 [a] | 4.50 [a] | 1.47 [b] | 4.50 [a] | 0.07 [b] |
| 3% | 47.10 [a] | 60.43 [a] | 4.40 [b] | 1.30 [c] | 2.20 [a] | 0.06 [b] |

* Figures in the same column followed by same letters are not significant.

### 3.1.5. Combined Impacts of Corncob Biochar and Drainage Water on Wheat Uptake of Heavy Metals

To determine the environmental impacts of biochar application coupled with irrigation with contaminated drainage water on the uptake of heavy metals into the food chain, samples of wheat tissues and grain were taken from the plants from each lysimeter at harvest. Data on heavy metal contents (expressed on a dry matter basis) of wheat parts are given in Table 9. The concentrations of Ni, Pb, and Cd in wheat grown on control lysimeters were significantly higher than concentrations in wheat grown on biochar incorporated or surface applied. This might be attributed to the increased solubility of heavy metals in the absence of biochar. The increased concentrations of Ni, Pb, and Cd in wheat grown on subsurface biochar application lysimeters were non-significantly higher than the concentrations in wheat grown on biochar incorporated into the soil. Results of the statistical analyses in all metal concentrations measured showed significant differences ($p \leq 0.05$) between biochar-amended lysimeters and the control with a greater effect of biochar rates than application methods.

**Table 9.** Heavy metal concentrations in wheat tissues and grains as affected by biochar and irrigation with Al-Moheet drainage water.

| Treatments Rate | Wheat Tissues [mg kg$^{-1}$] | | | Wheat Grains [mg kg$^{-1}$] | | |
|---|---|---|---|---|---|---|
| | Ni | Pb | Cd | Ni | Pb | Cd |
| **Biochar Incorporation** | | | | | | |
| 0% | 15.50 [a*] | 30.30 [b] | 4.60 [d] | 7.93 [a] | 20.87 [a] | 1.06 [a] |
| 1% | 10.50 [b] | 29.70 [ab] | 4.10 [d] | 7.50 [a] | 19.00 [a] | 0.13 [d] |
| 2% | 11.10 [b] | 28.90 [b] | 3.90 [e] | 8.40 [a] | 20.33 [a] | 0.15 [d] |
| 3% | 11.60 [ab] | 30.20 [ab] | 4.40 [b] | 7.90 [a] | 20.60 [a] | 0.19 [ab] |
| **Biochar Mulching** | | | | | | |
| 0% | 15.50 [a] | 32.30 [b] | 4.50 [d] | 8.63 [a] | 20.83 [a] | 1.06 [a] |
| 1% | 12.20 [ab] | 31.40 [ab] | 4.30 [c] | 8.52 [a] | 19.57 [a] | 0.14 [d] |
| 2% | 11.90 [ab] | 33.50 [a] | 4.50 [a] | 8.43 [a] | 19.50 [a] | 0.18 [c] |
| 3% | 12.10 [ab] | 31.20 [ab] | 4.40 [b] | 8.57 [a] | 20.30 [a] | 0.21 [b] |

* Figures in the same column followed by the same letters are not significant.

The heavy metal concentrations in wheat tissues decreased with an increasing biochar rate but showed no significant differences. The application of biochar significantly ($p \leq 0.05$) decreased the concentration of heavy metals in wheat tissues and grain compared to the control due to an improvement in soil properties and soil organic matter content with biochar application. Heavy metal concentrations in wheat tissues and grain increased at lower application rates of 1% and 2% with insignificant differences and then decreased or leveled at the higher rate of 3%. This indicated that wheat plants may have reached the maximum uptake and excess metals were adsorbed by biochar particles, resulting in a lower bioavailability of heavy metals. Moreover, the decrease in concentrations of heavy metals in wheat straw and grains at all application rates and with all methods of application could have been due to increased organic carbon in the soil.

Increases in heavy metal (Ni, Pb, and Cd) concentrations between biochar-amended lysimeters were relatively small, and they were insignificant in most cases with an increasing application rate. This indicated that the availability of Ni, Pb, and Cd to wheat plants was not dependent only on the concentration of elements in irrigation water and biochar rate or the method of application. Biochar affected other soil properties, such as Ca CO$_3$, content, changes in soil pH, CEC, moisture constants and organic matter, which may also induce important effects on heavy metals fate in the investigated sandy soil. Wheat tissues and grain in all biochar treatments had Ni, Pb, and Cd concentrations within the reported optimum ranges. The average Ni, Pb, and Cd concentrations were all within the optimum range conveyed in the literature for wheat in all biochar treatments and only increased above this range for untreated control if irrigation continued with such contaminated drainage water. In general, no toxicity symptoms were observed in any biochar treatments.

### 3.2. Incubation Experiment

3.2.1. Combined Effects of Corncob Biochar and Drainage Water on Microbial Biomass in a Sandy Soil

The results presented in Table 10 show a significant increase in the proportions of fungi and bacteria (F/B ratio) with increasing rates of biochar after 20 days of incubation despite irrigation with contaminated water compared to the control. Several researchers have indicated that the large-scale use of different types of biochar on a long-term basis has resulted in a significant increase in the proportion of soil fungi and bacteria with a preference for stimulating soil fungi over soil bacteria [37–39].

**Table 10.** Total count of bacteria (B) [$\times 10^6$ cfu g$^{-1}$] and fungi (F) [$\times 10^4$ cfu g$^{-1}$] and soil Fungal/Bacterial (F/B) ratio in sandy soil amended with corncob biochar irrigated with Al-Moheet drainage water.

| Treatments Rate | Days after Sandy Soil Incubation | | | | | | | | | | | |
| --- | --- | --- | --- | --- | --- | --- | --- | --- | --- | --- | --- | --- |
| | 5 | | | 10 | | | 15 | | | 20 | | |
| | F | B | F/B | F | B | F/B | F | B | F/B | F | B | F/B |
| **Biochar Incorporation** | | | | | | | | | | | | |
| 0% | 15 d* | 18 d | 0.83 | 52 cd | 48 cd | 1.08 | 92 d | 97 bc | 0.94 | 96 c | 95 c | 1.01 |
| 1% | 28 bc | 23 c | 1.20 | 61 c | 51 c | 1.19 | 113 c | 101 bc | 1.11 | 140 ab | 113 bc | 1.23 |
| 2% | 33 ab | 31 ab | 1.06 | 74 ab | 65 ab | 1.13 | 129 a | 109 ab | 1.18 | 143 ab | 129 ab | 1.11 |
| 3% | 35 a | 33 a | 1.06 | 79 a | 77 a | 1.02 | 128 ab | 119 a | 1.07 | 154 a | 135 a | 1.14 |
| **Biochar mulching** | | | | | | | | | | | | |
| 0% | 15 d | 18 c | 0.83 | 52 d | 48 d | 1.08 | 92 d | 97 d | 0.94 | 96 c | 95 c | 1.01 |
| 1% | 31 bc | 28 ab | 1.10 | 79 bc | 72 c | 1.09 | 146 c | 123 bc | 1.10 | 152 ab | 150 ab | 1.01 |
| 2% | 36 ab | 31 a | 1.16 | 85 b | 99 ab | 0.85 | 158 ab | 133 b | 1.18 | 153 ab | 154 a | 0.99 |
| 3% | 41 a | 30 ab | 1.36 | 104 a | 101 a | 1.03 | 159 a | 152 a | 1.04 | 158 a | 150 ab | 1.05 |

Inc., incorporation method, Mul., Mulching method, F, fungi, B, bacteria. * Figures in the same column followed by same letters are not significant.

These results are consistent with previous research by de Vries et al. [40], who reported that adding organic amendments to the soil increased the soil F/B ratio due to increased fungal growth. In contrast, Gomez et al. [41] reported a significantly lower ratio of fungi to bacteria in different biochar-amended soil types. By comparing fungi to bacteria, fungi can assimilate carbon sources more efficiently under large-scale biochar loads, and fungal hyphae grow in biochar pores using the most stable and biodegradable carbon sources [5,36,42].

3.2.2. Combined Effects of Corncob Biochar and Drainage Water on Soil Resistance Index (SRI)

In this study, the significant effect of adding biochar to soil on the total number of bacteria and fungi is presented, and its increased value was verified in SRI sandy soils despite irrigation with Al-Moheet wastewater (Table 11). Soil resistance index (SRI) values for soil microbial biomass were positive throughout the experiment; however, they differed depending on the metal type, biochar rate, and method of application. The lower SRI values indicated a stronger effect of the drainage irrigation water contaminated with nickel, lead, and cadmium on the soil microbial balance, which led to a decrease in microbial resistance. The sandy soil resistance index (SRI) increased to a higher extent as the biochar rate increased, whether incorporated or mulched. However, higher and significant SRI values for soil microbial biomass were observed in biochar-amended sandy soils where mulching indicated greater resistance to these metals' toxicity than incorporation into the soil. Cadmium (Cd) caused stronger disturbances for soil microorganisms irrigated with drainage water, albeit amended with corncob biochar. Haddad et al. [5] and Yin et al. [43] also demonstrated the sensitivity of soil microbial biomass to heavy metal contamination of the soil. Heavy metals bioaccumulating in the food chain of animals and humans due to increased utilization of food crops can increase the environmental toxicity risks associated with the remediation of contaminated soil with edible feed or plants [44,45].

**Table 11.** Soil Resistance Index (SRI) of bacteria and fungi to Cd, Ni, and Pb concentrations in drainage water for the irrigation of biochar-amended sandy soil.

| Treatments Rate | Biochar Incorporation | | | Biochar Mulching | | |
|---|---|---|---|---|---|---|
| | Soil Resistance Index (SRI) of Bacteria to Heavy Metals | | | | | |
| | Cd | Ni | Pb | Cd | Ni | Pb |
| 1% | 0.15 [b]* | 0.25 [c] | 0.24 [c] | 0.21 [c] | 0.38 [c] | 0.41 [c] |
| 2% | 0.26 [a] | 0.33 [b] | 0.34 [b] | 0.33 [a] | 0.43 [b] | 0.56 [b] |
| 3% | 0.26 [a] | 0.35 [a] | 0.47 [a] | 0.32 [b] | 0.51 [a] | 0.64 [a] |
| | Soil Resistance Index (SRI) of Fungi to Heavy Metals | | | | | |
| | Cd | Ni | Pb | Cd | Ni | Pb |
| 1% | 0.11 [c] | 0.26 [c] | 0.41 [c] | 0.16 [c] | 0.28 [c] | 0.45 [c] |
| 2% | 0.22 [b] | 0.45 [ab] | 0.57 [b] | 0.27 [ab] | 0.49 [a] | 0.59 [b] |
| 3% | 0.24 [a] | 0.42 [a] | 0.64 [a] | 0.28 [a] | 0.47 [b] | 0.67 [a] |

* Figures in the same column followed by same letters are not significant.

In general, the increased rates of biochar application caused a significant increase in the soil resistance index, and the surface-applied biochar caused highly significant changes compared to incorporation, which may be attributed to the biochar on the soil surface adsorbing more heavy metals reducing bioavailability for soil microbial biomass. High and optimal SRI values were observed in sandy soil treated with a higher rate of biochar at 3% indicating that sandy soils became more resistant to heavy metal toxicity. Cadmium (Cd) instigated stronger disturbances for soil microorganisms than nickel (Ni) and lead (Pb) in the sandy soil resulting in lower values of the soil resistance index (SRI). The soil resistance index (SRI) is an effective indicator to measure the soil's capability to continue functioning under environmental stress conditions such as irrigation with water contaminated with

heavy metals. Furthermore, the soil resistance index (SRI) is an effective indicator of soil microbial responses to external environmental stress [27].

3.2.3. Combined Effects of Corncob Biochar and Drainage Water on Soil Enzymatic Activities

Table 12 shows the results of corncob biochar application as a soil amendment on the arginase and urease activities in the soil. Urease affects nitrogen transformation in soil effectively and plays an important role in nitrogen release under different soil conditions. The results of this research indicated that, after 20 days of incubation, significant increases ($p \leq 0.05$) in arginase and urease activities were recorded in all rates of biochar-added soil, and 3% biochar was more effective than the others. It is well established that the most important indicators of soil health are microbial biomass and enzyme activity, which reflect the effects of different agricultural practices or pollution on soil health and also reflect the soil's ability to resist and purify itself from sources of pollution [5,9,46].

**Table 12.** Effect of irrigation with Al-Moheet drainage water on the activity of arginase (AR) and urease (UR) in sandy soil amended with corncob biochar.

| Treatments Rate | Biochar-Soil Incorporation Applied | | Biochar-Soil Mulching Applied | |
|---|---|---|---|---|
| | AR * | UR | AR | UR |
| **After 5 Days of Incubation** | | | | |
| 0% | 13.43 [d]** | 7.65 [d] | 15.11 [d] | 6.87 [d] |
| 1% | 26.78 [bc] | 15.44 [bc] | 27.78 [bc] | 13.96 [bc] |
| 2% | 28.43 [ab] | 16.38 [b] | 29.45 [ab] | 14.35 [b] |
| 3% | 32.33 [a] | 18.15 [a] | 32.78 [a] | 17.43 [a] |
| **After 10 Days of Incubation** | | | | |
| 0% | 15.44 [d] | 8.56 [d] | 16.16 [d] | 8.27 [d] |
| 1% | 36.78 [c] | 26.37 [bc] | 26.89 [bc] | 16.46 [bc] |
| 2% | 40.89 [ab] | 28.88 [b] | 30.56 [ab] | 18.91 [b] |
| 3% | 40.99 [a] | 35.96 [a] | 30.78 [a] | 24.88 [a] |
| **After 20 Days of Incubation** | | | | |
| 0% | 15.67 [d] | 9.61 [d] | 14.87 [d] | 9.59 [d] |
| 1% | 40.76 [bc] | 31.57 [bc] | 31.56 [bc] | 18.47 [bc] |
| 2% | 43.56 [b] | 30.27 [b] | 31.87 [b] | 19.69 [b] |
| 3% | 49.67 [a] | 39.88 [a] | 38.98 [a] | 25.35 [a] |

* AR—arginase [$\mu g\ NH_4$-N $g^{-1}\ h^{-1}$], UR—urease [$\mu mol\ NH_3$-N $g^{-1}\ h^{-1}$], ** Figures in the same column followed by same letters are not significant.

Adesina et al. [9] confirmed a decrease in the bioavailability of toxic soil heavy metals due to the enhancement of the soil microbial biomass, thus resulting in increased enzyme activity that improved soil health and fertility. For all biochar treatments in this incubation experiment, arginase and urease activity significantly increased regardless of irrigation with cadmium, nickel, and lead-contaminated drainage water, compared to the control. In addition, biochar application in sandy soils may alleviate heavy metal levels on microbial biomass affecting soil enzyme activity of arginase and urease. Incorporating or spreading biochar over sandy soil can lead to precipitating, immobilizing, or stabilizing heavy metals in the soil through precipitation, adsorption, stabilization, or organic complexing, thus reducing the availability of toxic metals for plants in sandy soil contaminated with irrigation water. At the time of overlapping incubation, biochar has proven to be an effective soil amendment to improve the health of sandy soils, reflected in the biochemical and biological properties of the improved soil. Ippolito et al. [47] reported that the addition of biochar to the soil was able to reduce the bioavailability of Ni, Cd, and Pb due to the increased microbial biomass in the sandy soils examined, while the responses of different enzymes to biochar and/or heavy metals impacts showed significant differences. It can be concluded that adding corncob biochar. either through soil incorporation or placing it on the soil

surface, during wheat or incubation experiments improved sandy soil productivity and soil health.

In conclusion, through corncob biochar application to sandy soil, it was possible to overcome the contamination effects of irrigation with Al-Moheet drainage water contaminated with heavy metals on sandy soil microbial and enzymatic activities.

This research was conducted to address soil health status after the addition of corncob biochar to sandy soil irrigated with Al-Moheet drainage water under desert sandy soil conditions. Under arid conditions, agricultural production is accompanied by harsh difficulties such as water scarcity, salinity build-up, soil infertility, desertification, and consequently, low crop yield. These severe difficulties can be adequately addressed using biochar treatment of soil and irrigation water. Biochar soil application or water treatment has become the focus of soil and water research compared to other physical and chemical methods of remediation. Over the past three decades, all conducted research has indicated the positive role of biochar application in soil and water remediation to provide adequate irrigation water delivery, increase soil nutrient availability, reduce the availability of heavy metals in soil, and increase crop yield without causing environmental mutilation [4,5,8,48]. In Egypt, massive amounts of organic waste have been generated due to intensive agricultural activities to produce food for a growing population and attempts at recycling must be developed with minimal environmental risks [4].

In Egypt, soil contamination with heavy metals due to irrigation with polluted unconventional water resources has become a major concern due to its non-degradable nature and bioavailability for uptake by plants and organisms, thus heavy metals are susceptible to entering agricultural food products [4,36]. The accumulation of heavy metals such as Ni, Pb, and Cd in soil is of increasing concern due to their lethal and numerous dysfunctional effects on soil microorganisms [14]. Contaminated unconventional irrigation water resources with heavy metals can cause many biochemical changes in soils and plants resulting in reduced growth and yields leading to national food insecurity. In this research, biochar has been applied to soil as an innovative carbon-rich material for heavy metal processing as biochar has the ability to create complexes with heavy metals present in irrigation water and contaminated soil, thus reducing their bioavailability [5,16].

Extensive agricultural areas in Egypt have arid and semi-arid conditions and severe problems of salinization because of irrigation with low-quality water along with poor drainage infrastructures and low soil fertility or nutrient availability. In this regard, irrigation with saline water is compelling farmers in arid areas to devise innovative technologies to preserve crop yield and quality while adapting to the degradation of natural resources [49,50]. Low production costs, the high cation exchange capacity, pH, surface functional groups, and porous structures are some of the intrinsic properties of corncob biochar making it an ideal choice as an adsorbent for heavy metal and a possible applicant for long-term carbon storage and sequestration [10].

Sawyer et al. [51] stated that heavy metals in public water supplies are usually arbitrarily defined as those for which drinking water standards are generally in the range of 1 mg L$^{-1}$ or less. The maximum contaminant level of drinking water for the selected elements Ni, Pb, and Cd is 0.1, 0.1, and 0.005 mg L$^{-1}$, according to USEPA [23]. In the absence of organic residues or biochar, leaching could be a serious problem in sandy soils in arid areas with its inherited low surface area, low fertility, coarse texture, and very poor organic matter content and water-holding capacity. Biochar is a carbon-rich product, and its potential to improve soil fertility and decrease the bioavailability and leachability of a range of heavy metal contaminants within soils has been recognized globally [4,5]. In particular, sandy soil has a limited specific surface area (0.01–0.1 m$^2$ g$^{-1}$) compared to clay soil (5–750 m$^2$ g$^{-1}$), thus their water pollutant or nutrient retention capacities are low [52].

The central quality of biochar as a soil conditioner is its highly porous structure, which improves water retention and increases soil surface area [4,49,52,53]. A study by Liang et al. [54] on sandy soil biochar inclusion demonstrated specific surface area and soil CEC increases relative to untreated soil, since the ability of biochar to adsorb or transport

nutrients is a crucial factor for nutrients to be generally retained in soils [5,8,34]. Since biochar has a higher surface area, greater negative surface charge, and greater charge density than other soil organic matter, it has the capacity to adsorb cations to a greater extent. Additionally, biochar has been shown to contribute directly to nutrient uptake, thus reducing nutrient leaching or pollutants, thus increasing nutrient use efficiency with higher yield production and therefore a significant correlation between extractable heavy metals and total applied heavy metals is unlikely [16,34,55].

In Egypt under arid and semi-arid conditions, due to the alkaline and calcareous nature of desert soils and low precipitation, no increases in heavy metals in the subsoil or groundwater are expected. In addition, the water table is generally deep and the potential for heavy metals to reach the groundwater is low where biochar or organic manure is used. Therefore, if by accident due to flood irrigation the sandy soil moisture increased above saturation, the hypothesis inferred is that the effect of high rates of biochar application to sandy soils would decrease the concentrations of selected heavy metals in groundwater compared to untreated sandy soils. It is clear that the most important soil characteristics influencing soil-metal mobility are pH, organic matter quality and quantity, Fe and Mn oxides, and the percent of clay content [56,57].

Finally, under the conditions of this study, the hypothesis that heavy metals can accumulate in the surface layer of the sandy soil under study, leak into groundwater, or uptake into cultivated plants to a dangerous level in the short term must be rejected. An integrated approach of using biochar under field conditions with optimal irrigation and fertilization can help achieve soil remediation as well as soil health to improve yields, but most research is in its infancy. The properties of biochar are highly dependent on the raw materials and production process, and extensive research is also necessary for standardization and regulation in this regard. The interactions between biochar and microorganisms in different soils contaminated with heavy metals are still poorly understood, since the application of biochar in soil is a non-reversible approach, so systematic monitoring is necessary from the perspective of soil health, human health, and the environment, as well as food safety. Although ongoing research is providing new insight into the application of biochar for the uptake of heavy metals from soil, data are still insufficient for its use in large-scale applications under field conditions of arid regions.

## 4. Conclusions

The research results revealed that insignificant and small amounts of heavy metals at all biochar application rates reside in the water-soluble form, which is possibly available for plant uptake and or more vulnerable to leaching than other soil–biochar mixture adsorbed fractions. Overall, this study indicated that heavy metals applied in drainage irrigation water to sandy soil treated with biochar rarely migrate to any depth in the soil and very little reach the groundwater via the process of intensive flood irrigation. Heavy metal concentrations after the final irrigation event and wheat harvest were within permissible limits in soil leachate and wheat grain concentrations of nickel, cadmium, and lead were far below concentrations considered phytotoxic to humans. However, in the long run, irrigation with Al-Moheet drainage water should be monitored regularly, or irrigation with such saline and contaminated drainage water should even be stopped despite repeated applications of organic matter or biochar. It is necessary to confirm these results in the long term by conducting field studies on land irrigated with these waters over a long duration. It should be borne in mind that irrigating sandy soils with highly saline drainage water and high heavy metal content will be a discontinuous process rather than an annual or even seasonal practice that will constantly add salts and heavy metals to such soil, which may destroy soil health.

**Author Contributions:** Conceptualization, M.M.A.E.-A., J.L. and S.A.H.; data curation, M.M.A.E.-A., A.M.M. and M.M.A.E.-M.; formal analysis, A.M.M. and M.M.A.E.-M.; funding acquisition, M.M.A.E.-A.; investigation, M.M.A.E.-M. and S.A.H.; methodology, A.M.M., M.M.A.E.-M. and S.A.H.; project ad-ministration, M.M.A.E.-A.; resources, S.A.H.; software, M.M.A.E.-M.; supervision, M.M.A.E.-A. and S.A.H.; validation, A.M.M. and J.L.; visualization, J.L. and S.A.H.; writing—original draft, M.M.A.E.-A., A.M.M., M.M.A.E.-M. and S.A.H.; writing—review and editing, J.L. and S.A.H. All authors have read and agreed to the published version of the manuscript.

**Funding:** This research received no external funding.

**Institutional Review Board Statement:** Not applicable.

**Data Availability Statement:** Data sharing not applicable.

**Acknowledgments:** The authors would like to thank the Faculty of Agriculture, Minia University, for their support in this research work.

**Conflicts of Interest:** The authors declare no conflict of interest.

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
