# Peer review of "Wheat Crop Yield and Changes in Soil Biological and Heavy Metals Status in a Sandy Soil Amended with Biochar and Irrigated with Drainage Water"

_agriculture, doi:10.3390/agriculture12101723_

Round 1
Reviewer 1 Report
Reviewer
MDPI – Agriculture
Manuscript Number: agriculture-1956855
Title: « Temporary soil health status in biochar-amended sandy soil irrigated with drainage water».
The topic studied in this article is very relevant at the present time. The use of corn biochar at 3% has been found to temporarily improve the environment and health of sandy soil despite irrigation with drainage water.
Despite this, there are a number of comments and suggestions.
line 19 Remove repetitions of the "%" sign
line 100 You need to add the purpose of the study and tasks.
line 125 Is it degrees: "300℃"?
line 226 Did you mean urease activity, not urase?
The conclusion is very long and needs to be shortened. According to the purpose of the study and the tasks, it is necessary to give concise answers to the solution of these problems. The conclusion should not continue the discussion of the material, but only concrete results should be presented.

Author Response
(Reviewer 1):
- line 19 Remove repetitions of the "%" sign.
Revised.
- line 100 You need to add the purpose of the study and tasks.
Thank you for your comment, please see Lines (90-98) the objectives of the study.
- line 125 Is it degrees: "300℃"?.
Yes, revised.
- line 226 Did you mean urease activity, not urase?
Yes, revised.
- The conclusion is very long and needs to be shortened. According to the purpose of the study and the tasks, it is necessary to give concise answers to the solution of these problems. The conclusion should not continue the discussion of the material, but only concrete results should be presented.
Thank you very much for your comment, the conclusion we shortened and revised.
Finally, thank you very much for your time and effort helping us improving our manuscript. Hopefully the corrections required have been adequately made.

Reviewer 2 Report
The manuscript by Mohieyeddin M. Abed El-Azeim et al. presented a study on how biochar addition affect health status of soil irrigated with drainage water. They found high feasibility of corncob biochar application at a rate of 3% to improve the environment and the health of sandy soil despite
irrigation with drainage water. The study is interesting and worth to publish once all my concerns as indicated in attachement be responsed in future revisions. My main comments including the following aspects:
1. The title is not so specific, instead, the broad words using in the present title make it difficult for readers to identify what were studied by the paper.
2. The data analysis method is OK BUT NOT CLEARLY STATED, and all the tables with multiple comparisions missed the results of ANOVA, which is not rational. Please include in the revision.
3. The citations are in bold while some aother word are also in bold. Please check if this is required by the journal.
4. Some of the format need to be revised, please see comments in attached.
5. The study have lot of data. But most of them are isolateing for each other, e.g., the micrbial data and the soil data, if possible, please analye all data together, using methods like SEM or so, to explore more new findings. If not, I suggest to remove some of the tables, focusing on the key findings of the study.
Other sugesstions could be found in the attachment.

Author Response
(Reviewer 2):
Thank you very much for sending us very detailed reviews of the Manuscript.
- The title is not so specific, instead, the broad words using in the present title make it difficult for readers to identify what were studied by the paper.
Changed to: Changes in soil biological status in biochar-amended sandy soil irrigated with drainage water.
- The data analysis method is OK BUT NOT CLEARLY STATED, and all the tables with multiple comparisons missed the results of ANOVA, which is not rational. Please include in the revision.
Thank you very much for your comment, completely agree. But here, analysis of variance showed significant differences among the treatments, so the Duncan test was preferred and performed, we have a lot of data and tables, so we thought this is an appropriate way to present our work.
- The citations are in bold while some other words are also in bold. Please check if this is required by the journal.
Yes, it was a mistake, revised in the whole manuscript.
- Some of the format need to be revised, please see comments in attached.
Thank you very much, all formats were revised according to your suggestions.
Finally, thank you very much for your time and effort in helping us improve our manuscript.
We did all of your suggested points listed in your report or attached file as much as we can Hopefully, the corrections required have been adequately made.

Reviewer 3 Report
The manuscript needs thorough revision. I have made comments, corrections, and suggestions on manuscript's pdf file. Besides those following comments are also provided for revision. Hopefully it will be revised and improved.
General Comments
1. Many sentences are too long, and their comprehension is not clear for example” surface flood irrigation is the predominant irrigation……” line No. 42-45 introduction section.
2. Language of the paper is not clear, in many places the selection of words makes the sentences non-comprehensible.
3. Many paragraphs do not have a sequence of events i.e. the statements or sentences are irregularly placed in a paragraph making comprehension difficult.
4. The article’s general structure and composition is well but the way in which information is presented is quite difficult to understand. The text needs thorough language revision.
5. I can’t understand whether the control treatment for both mulching and incorporated application is the same?? Because in many tables values for control in both application methods are the same (Table 4 and 8) but in other tables (for example table 7,9) the values for control of mulching and incorporation are different??
6. The regression analysis should have been applied to make the results clearer and more robust.
7. Many paragraphs added after the enzyme activity, are irrelevant and have no relation to the study presented in this paper.
Section wise comments
Materials and methods
1. How can a lysimeter be established with sand?? Since sand is porous and cannot work as a barrier between two treatment variables.
2. What is the exact size of the Lysimeter?? How you have calculated the dose of biochar if the size of the lysimeter is unknown?
3. Wheat cultivation process, name of variety, and a number of irrigations must be mentioned in the methods.
4. Figure 1 shows only one magnification which is 500 while it is claimed in the statement that different magnifications were used in the analysis.
5. What material was used to construct the lysimeters?? Because earlier it has been said that lysimeters were established in newly reclaimed sand..
6. The design of the experiment is missing, how many replications are applied, what is the statistical design and what factors should be included in the methods.
7. The design of the lysimeter is ambiguous. I suggest that a schematic drawing or photograph of the lysimeter may be included for clear understanding. It is unclear that how the leachate has been collected while there is no mention of a draining line or source for leachate collection.
8. Al-Moheet drain is 135 km in length so its pollution load must be different at different places. I suggest you must include a map of the drain showing your experimental site.
9. In the incubation experiment, was the post-harvest soil added with fresh biochar material?
10. I believe instead of a separate incubation study it could be better to analyze soil from the root zone of the wheat crop for microbial and enzymatic assessment because it would have produced soil-plant-microbe interaction in a quite better way. Here the authors have set up a separate experiment for microbial and enzyme activity analysis which lacks plant interaction.
11. The methods for soil enzyme activity analysis must be explained in detail.
12. Use the same units for cations in soil, biochar, and water.
13. Conclusion is too long it should be short and objective/hypotheses specific
Results and Discussion
Evaluation of Al-Moheet drainage water quality for irrigation
1. The water quality parameters or salinity levels are not present in table 04 as claimed in the text in the mentioned section.
2. Although, EC has increased significantly from the initial soil analysis, but yet it did not cross the threshold level of salinity (> 4 dSm-1).
3. Results on heavy metals in water quality are missing in this section which is one of the objectives of the study.
Combined impacts of corncob biochar and drainage water on wheat productivity and quality characteristics
1. Results are not sufficiently described.
2. Table 4 shows non-significant results between the two application methods, while it has been claimed as significant by the authors.
3. As mentioned in the previous section, the Al-Moheet drainage water is not suitable for crops as its salinity level is high. How it enhanced crop parameters significantly.
Combined impacts of corncob biochar and drainage water on some Physico-chemical characteristics
1. pH has not been increased (Table 5) as claimed by the authors in the text.
2. If the drainage water has high EC (categorized as severe as claimed by you) then why the EC in soil has not been increased after many irrigations? Even the control soil EC is too below the salinity threshold level. Results must be verified.
3. It is interesting that the organic matter in soil has significantly reduced despite application of Biochar (Table 5)
4. No units for chemical parameters (Table 5).
Combined impacts of corncob biochar and drainage water on lysimeters soil leachate composition
1. Leachate is not a suitable word it should be replaced with accumulation at lower profile, because leachate is the liquid that drains from the soil while in the study it is nowhere mentioned that the liquid drained water has been collected and analyzed. Therefore, word leachate is confusing.
2. Line 296-298, the table 6 does not include the metal concentrations in top soil, their rate of movement in sub-soil, and uptake by wheat plants. Please be specific in your presentation of results.
3. Please add units of measurement in Table 6.
4. The leachate results are very low even in control. It means much of the metals have been adsorbed in soil by biochar but what about the control in which no biochar is applied…This is not scientifically justified.
5. Table 6 and 8 are confusing.
Combined impacts of corncob biochar and drainage water on soil heavy metals accumulation and movements
1. There is no heavy metal movement data in table 7 or 8.
2. Please explain the calculation of total heavy metals…why they are in such huge concentration than the extracted heavy metals.

Author Response
(Reviewer 3):
Thank you very much for sending us very detailed reviews of this Manuscript, we tried as much as we can to do all of your valuable comments and corrections even in your attached file or report.
Abstract:
Revised and modified as suggested in your attached file.
Materials and methods
- How can a lysimeter be established with sand?? Since sand is porous and cannot work as a barrier between two treatment variables.
Thank you for your comments, we added Fig 2 and 3 for more clarification.
- What is the exact size of the Lysimeter?? How have you calculated the dose of biochar if the size of the lysimeter is unknown?
Please see the Figure 2. Lysimeter experiment construction (added)
- Wheat cultivation process, name of variety, and a number of irrigations must be mentioned in the methods.
Revised
- Figure 1 shows only one magnification which is 500 while it is claimed in the statement that different magnifications were used in the analysis.
Modified and Revised
- What material was used to construct the lysimeters?? Because earlier it has been said that lysimeters were established in newly reclaimed sand.
Please see the Figure 2. Lysimeter experiment construction (added)
- The design of the experiment is missing, how many replications are applied, what is the statistical design and what factors should be included in the methods.
Please see : 1.1.2 Lysimeter experiment construction
- The design of the lysimeter is ambiguous. I suggest that a schematic drawing or photograph of the lysimeter may be included for clear understanding. It is unclear that how the leachate has been collected while there is no mention of a draining line or source for leachate collection.
Schematic drawing and photograph added
- Al-Moheet drain is 135 km in length so its pollution load must be different at different places. I suggest you must include a map of the drain showing your experimental site.
Map added (Figure 3)
- In the incubation experiment, was the post-harvest soil added with fresh biochar material?
No
- I believe instead of a separate incubation study it could be better to analyze soil from the root zone of the wheat crop for microbial and enzymatic assessment because it would have produced soil-plant-microbe interaction in a quite better way. Here the authors have set up a separate experiment for microbial and enzyme activity analysis which lacks plant interaction.
Thank you very much for this important comment, but after post-harvest, we took 2 kg from each lysimeter without biochar addition to measure microbial activity under controlled conditions.
- The methods for soil enzyme activity analysis must be explained in detail.
Thank you very much, but all information needed can be found clearly in the citation and we think this preferred for concise the manuscript.
- Use the same units for cations in soil, biochar, and water.
Different units of soil analysis for cations are completely different from water analysis.13. Conclusion is too long it should be short and objective/hypotheses specific
Shortened and revised
Results and Discussion
Evaluation of Al-Moheet drainage water quality for irrigation
- The water quality parameters or salinity levels are not present in table 04 as claimed in the text in the mentioned section.
Yes, this was mistake , revised.
- Although, EC has increased significantly from the initial soil analysis, but yet it did not cross the threshold level of salinity (> 4 dSm-1).
In this experiment, we advise the farmers to stop irrigate with such water to stop soil from reaching the hazard salinity level on the long run.
- Results on heavy metals in water quality are missing in this section which is one of the objectives of the study.
Revised
Combined impacts of corncob biochar and drainage water on wheat productivity and quality characteristics
- Table 4 shows non-significant results between the two application methods, while it has been claimed as significant by the authors.
Presented data in Table 4 showed that studied wheat productivity and quality parameters were significantly affected by biochar addition rates. Revised.
- As mentioned in the previous section, the Al-Moheet drainage water is not suitable for crops as its salinity level is high. How it enhanced crop parameters significantly.
Although the high level of Al-Moheet drain water salinity it doesn’t increase the soil EC level above soil salinity 4 dSm-1
Combined impacts of corncob biochar and drainage water on some Physico-chemical characteristics
- pH has not been increased (Table 5) as claimed by the authors in the text.
Revised
- If the drainage water has high EC (categorized as severe as claimed by you) then why the EC in soil has not been increased after many irrigations? Even the control soil EC is too below the salinity threshold level. Results must be verified.
In this experiment, we advise the farmers to stop irrigate with such water to stop soil from reaching the hazard salinity level on the long run.
- It is interesting that the organic matter in soil has significantly reduced despite application of Biochar (Table 5)
Organic matter in soil has significantly increased (Table 5)
- No units for chemical parameters (Table 5).
Please see captions under the table.
Combined impacts of corncob biochar and drainage water on lysimeters soil leachate composition
- Leachate is not a suitable word it should be replaced with accumulation at lower profile, because leachate is the liquid that drains from the soil while in the study it is nowhere mentioned that the liquid drained water has been collected and analyzed. Therefore, word leachate is confusing.
Not confusing.
- Line 296-298, the table 6 does not include the metal concentrations in top soil, their rate of movement in sub-soil, and uptake by wheat plants. Please be specific in your presentation of results.
Revised
- Please add units of measurement in Table 6.
Elements concentrations in leachate fractions mg L-1
- The leachate results are very low even in control. It means much of the metals have been adsorbed in soil by biochar but what about the control in which no biochar is applied…This is not scientifically justified.
- Table 6 and 8 are confusing.
Table 6 indicating concentrations of heavy metals in soil leachate but Table 8 indicating heavy metal concentrations under 20 cm of soil layer.
Combined impacts of corncob biochar and drainage water on soil heavy metals accumulation and movements
- There is no heavy metal movement data in table 7 or 8.
This true.
- Please explain the calculation of total heavy metals…why they are in such huge concentration than the extracted heavy metals.
For sure total heavy metals are more than extracted ones

Round 2
Reviewer 2 Report
Most of my concerns have been addressed by the first revision.
However, I have five general comments, while there are only four responses. Please reconsider the Fifth one attached below, whether or not it could be modified.
Thanks,
************************
5. The study have lot of data. But most of them are isolateing for each other, e.g., the micrbial data and the soil data, if possible, please analye all data together, using methods like SEM or so, to explore more new findings. If not, I suggest to remove some of the tables, focusing on the key findings of the study.
Author Response
(Reviewer 2):
- Most of my concerns have been addressed by the first revision. However, I have five general comments, while there are only four responses. Please reconsider the Fifth one attached below, whether or not it could be modified.
- The study have lot of data. But most of them are isolateing for each other, e.g., the microbial data and the soil data, if possible, please analyze all data together, using methods like SEM or so, to explore more new findings. If not, I suggest to remove some of the tables, focusing on the key findings of the study.
Thank you very much for your time and effort in helping to improve our paper.
We sincerely apologize for this unintended error, at this time, it is difficult to modify it with time constraints and a large number of tables presented. this may be affected the manuscript's sequencing after reviewers corrections.
Finally, thank you very much for your time and effort in helping us improve our manuscript.
Reviewer 3 Report
The manuscript has been well improved in this revised version. But its quality can be improved more if the authors are serious to incorporate or address the issues raised by the reviewers. In the responses few of the answers are unsatisfactory for example a quarry that Table 7-8 does not possess movement results, the authors' response is just "this is true". There is no justification or any revision in the tables or caption. I suggest that the caption for these tables may be changed according to the results presented in tables. Also, the paragraphs starting from line No. 546 (reviewer v 1) to 590, line 603 to 634 have sticky notes. These have not been addressed or responded satisfactorily.

Author Response
(Reviewer 3):
- The manuscript has been well improved in this revised version. But its quality can be improved more if the authors are serious to incorporate or address the issues raised by the reviewers. In the responses few of the answers are unsatisfactory for example a quarry that Table 7-8 does not possess movement results, the authors' response is just "this is true". There is no justification or any revision in the tables or caption. I suggest that the caption for these tables may be changed according to the results presented in tables.
Revised and improved
- Also, the paragraphs starting from line No. 546 (reviewer v 1) to 590, line 603 to 634 have sticky notes. These have not been addressed or responded satisfactorily.
Thank you for your comments, we went through those lines again (line No. 546 (reviewer v 1) to 590, line 603 to 634) even in reviewer V1 or V2 and we didn’t find any comments suggested by reviewers.
We did all comment suggestions in your attached files as well.
Thank you very much for sending us very detailed reviews of this Manuscript. We appreciate your time and efforts in helping us to improve our manuscripts to appear in this very good shape.